# Prediction of pH Value of Aqueous Acidic and Basic Deep Eutectic Solvent Using COSMO-RS σ Profiles’ Molecular Descriptors

**DOI:** 10.3390/molecules27144489

**Published:** 2022-07-13

**Authors:** Manuela Panić, Mia Radović, Marina Cvjetko Bubalo, Kristina Radošević, Marko Rogošić, João A. P. Coutinho, Ivana Radojčić Redovniković, Ana Jurinjak Tušek

**Affiliations:** 1Faculty of Food Technology and Biotechnology, University of Zagreb, Pierottijeva Ulica 6, 10000 Zagreb, Croatia; mpanic@pbf.hr (M.P.); mradovic@pbf.hr (M.R.); mcvjetko@pbf.hr (M.C.B.); krado@pbf.hr (K.R.); ana.tusek.jurinjak@pbf.unizg.hr (A.J.T.); 2Faculty of Chemical Engineering and Technology, University of Zagreb, Marulićev Trg 19, 10000 Zagreb, Croatia; mrogosic@fkit.hr; 3CICECO—Aveiro Institute of Materials, Department of Chemistry, University of Aveiro, 3810-193 Aveiro, Portugal; jcoutinho@ua.pt

**Keywords:** artificial neural networks, COSMO-RS, deep eutectic solvents, multiple linear regression, piecewise linear regression

## Abstract

The aim of this work was to develop a simple and easy-to-apply model to predict the pH values of deep eutectic solvents (DESs) over a wide range of pH values that can be used in daily work. For this purpose, the pH values of 38 different DESs were measured (ranging from 0.36 to 9.31) and mathematically interpreted. To develop mathematical models, DESs were first numerically described using σ profiles generated with the COSMOtherm software. After the DESs’ description, the following models were used: (i) multiple linear regression (MLR), (ii) piecewise linear regression (PLR), and (iii) artificial neural networks (ANNs) to link the experimental values with the descriptors. Both PLR and ANN were found to be applicable to predict the pH values of DESs with a very high goodness of fit (*R*^2^_independent validation_ > 0.8600). Due to the good mathematical correlation of the experimental and predicted values, the σ profile generated with COSMOtherm could be used as a DES molecular descriptor for the prediction of their pH values.

## 1. Introduction

Green chemistry presents a way of creating and applying chemical products and processes that reduce or eliminate the use or production of substances that are hazardous to human health and the environment [1]. A growing area of research in green technology development is devoted to the design of new, more environmentally friendly solvents whose use would meet technological and economic requirements. Requirements for alternative solvents include a reasonable price, non-toxicity to humans and the environment, non-flammability, biodegradability, and possibility of regeneration or recovery [2,3]. Currently, known green solvents are water, carbon dioxide, bio-solvents, ionic liquids, and deep eutectic solvents. In the last decade, deep eutectic solvents (DESs) have received enormous attention in the academic community and the number of articles published has increased exponentially.

DESs were first described by Abbott et al. in 2003 as a mixture of a hydrogen bond donor (HBD) with a hydrogen bond acceptor (HBA), which exhibited much lower melting points than the pure compounds due to the formation of hydrogen bonds between constituent compounds [4,5,6]. Lately, DESs have shown great potential for industrial application thanks to their acceptable costs, the versatility of their physicochemical properties, and simple preparation. They also often present low cytotoxicity and good biodegradability. The properties that have gained them the environmentally friendly label are low volatility (reduced air pollution), nonflammability (process safety), and stability (potential for recycling and reuse). The number of structural combinations encompassed by DESs is tremendous; thus, it is possible to design DESs with unique physicochemical properties for a particular purpose. The physicochemical properties, such as the viscosity, density, and pH value, of DESs are crucial for industrial application of these solvents in terms of equipment materials, mass transfer, filtration, or pumping [7].

The pH values of aqueous solutions affect the enzyme activity, extraction efficiency, and stability of biologically active molecules. As such, the pH value is an important property of a solvent and, especially for DES design, one of the critical parameters. Though several papers have analyzed the pH behavior of DESs, there are still gaps in the understanding of how DES-forming compounds influence its pH value [8,9]. Despite this, some general conclusions can be outlined. For example, DESs containing organic acids (i.e., malic acid or oxalic acid) are, as expected, more acidic than those containing polyalcohols or sugars. The role of the water content in DESs regarding the pH behavior is still not entirely clear; however, it was observed that an increase in pH values with an increasing water content was reported for DESs with extremely low pH values while the pH values of DESs with pH in the higher range of values (lower acidity region) decreased with an increasing water content [7].

So far, the search for an ideal DES for a particular system has been guided by an empirical trial-and-error approach, with no systematic research into the structure–activity of DESs. Therefore, the rational design of these solvents for specific purposes is still in its infancy. Data collection on the application properties of DESs and the development of mathematical methods as a tool for the design of novel solvents are imperative for the industrial application of these solvents. The Conductor-like Screening Model for Real Solvents (COSMO-RS) is an ab initio computational method that may be used for the generation of the σ profile of a molecule. The σ profile shows the probability of finding surface segments with σ polarity on the surface of the molecule and contains the most relevant chemical information needed to predict the compound’s electrostatic, hydrogen bonding, and dispersion interactions [10]. The distribution of the charge, the width, and the height of the peaks in the σ profile vary with the nature of the molecules. Therefore, any change in the molecular structure can be quantified. By coupling the σ profile of DES-forming compounds with experimental data using model-generating methods such as multiple linear regression (MLR), piecewise linear regression (PLR), or artificial neural networks (ANNs), models for the description of DESs’ physicochemical properties can be developed [11,12,13,14]. In most studies, good model fitting of the literature viscosity, density, and pH values of the DESs was obtained [12,13]. The results showed that simple linear models such as MLR and more complex ones such as ANN could be used efficiently to predict the physical properties of specific DES groups (e.g., amine or sugar-based DESs), whereas it was difficult to create a single model covering the whole range of possible DES systems [11]. Commonly, simple mathematical models such as MLR were good enough for viscosity and density prediction while in the case of the pH value, more complex ANN models had to be used [11,13,15].

In this work, we report a model for the prediction of the pH values of acidic and basic DESs. For this purpose, the experimental pH values of 38 different DESs were evaluated, described, and mathematically interpreted. For the development of mathematical models, DESs were firstly numerically described using σ profiles estimated by the COSMOtherm software. After the description of DESs, the following models were used: (i) MLR, (ii) PLR, and (iii) ANN to link the experimental values with the descriptors. In the end, the prepared models were statistically verified.

## 2. Results and Discussion

### 2.1. DES Characteristics: Experimental pH Values and σ Profiles

This work aimed to develop a simple and robust mathematical model for predicting the pH values of DESs based on *S*^i^_mix_ descriptors. To develop a user-friendly model to predict pH values in the wide range, we selected both acidic and basic DESs from our database. We chose 38 DESs by carefully selecting and varying different HBA, HBD, and water shares (Table 1). Selected HBAs and HBDs can be roughly classified as quaternary ammonium salts (choline chloride, betaine), amino acids (proline), organic acids (citric and malic acid), and sugars (fructose, glucose, sucrose, xylose). In comparison to HBA, there are more HBD candidates from previously mentioned classes and it has been shown that they have an immediate effect on pH values (Table 1). Overall, all synthesized DESs cover a wide range of pH values from 0.36 for Ch:CA containing 30% water (*w*/*w*) to 9.31 for Ch:U containing 10% water (*w*/*w*). Monitoring the pH values of the same HBA/HBD pair while varying the DES water content shows that water influences the measured pH value. However, this influence is a distinctive characteristic of an individual DES and cannot be extended to all DESs studied in this work.

Furthermore, DESs were mathematically described using the σ profile defined with the COSMOtherm software. The HBA and HBD molecules were optimized in TmoleX, both from an energy and geometry point of view. The generated COSMO files contain all information necessary for the calculation of the σ profile function and thus for the calculation of the σ profile descriptors. For the preparation of the descriptor set, the DESs were modeled as a molar mixture of HBA and HBD according to Table 1. The σ profile curves for each HBA and HBD were divided into 10 regions, the area under each region was calculated, and their numerical values were correlated with the experimental pH values using mathematical models.

### 2.2. Multiple Linear Regression and Piecewise Linear Regression

The assessment of the MLR and PLR model applicability to predict the pH values of DESs was based on the correlation coefficient values, *R*^2^, *R*^2^_adj_, and *RMSE*. The obtained model coefficient values and the basic statistical analysis are presented in Table 2 while a comparison between the experimental and model-estimated pH values is given in Figure 1.

As described in the literature, linear regression calculates an equation that minimizes the distance between the fitted line and all data points. In general, a model fits the data well if the discrepancies between the observed and predicted value are minimal and unbiased. According to Cheng et al. (2014) [16], the coefficient of determination and adjusted coefficient of determination can be considered as summary measures for the goodness of fit of any linear regression model. Moreover, Le Mann et al. (2010) stated that the model can be regarded as appropriate if the coefficient of determination is above 0.75 [17]. Based on this, it can be concluded that both the MLR (*R*^2^ = 0.7758) and PLR (*R*^2^ = 0.9654) models developed in this work are applicable for the description of DESs’ pH values based on *S*^i^_mix_ descriptors but not with the same accuracy. When analyzing *RMSE* errors, it is evident that the PLR model (Figure 1b) ensures significantly smaller data dispersion (*RMSE* = 0.6558) in comparison to the MLR model (*RMSE* = 1.1865) (Figure 1a). As previously described, a high-accuracy model is strongly desired. However, the increase in the accuracy is usually accomplished by the increase in the complexity of the models by increasing the number of model parameters. For practical application, a model with fewer parameters is easier to interpret and, therefore, more suitable for the application.

A high *R*^2^ value alone does not guarantee that the model fits the data well, so the model’s goodness of fit was further confirmed by residual analysis. The residuals from a fitted model are the differences between the responses observed and the corresponding prediction of the response computed using the regression function. If the model’s fit to the data was correct, the residuals would approximate the random errors that make the relationship between the explanatory variables and the response variable a statistical relationship. Therefore, if the residuals appear to behave randomly, it would suggest that the model fits the data well [18]. Analyzing the results presented in Figure 2, the residuals for the MPLR and PLR models were found to be normally distributed (Figure 2a,b). Furthermore, because the residual plots were gathered roughly along a straight line, the normality condition was met. The bell-shaped histograms that display the measurement distribution also verified the normal distribution of the residuals (Figure 2a,b). The residual vs. predicted value plots (Figure 2a,b) reveal that the residuals have no pattern, implying that the models match the experimental data well. Additionally, the residuals were found to range around the central value (Figure 2a,b) without obvious outliers, which means that the level of randomization was appropriate and that the sequence of testing had no effect on the findings [19].

Analysis of the MLR and PLR model coefficients showed that all coefficients, except *b*_6_ (coefficient multiplying *S*^6^_mix_), were statistically significant. It can also be noticed that for both models, the coefficients from *b*_1_ to *b*_5_ have a positive influence on the output variable while the coefficients from *b*_6_ to *b*_10_ have a negative influence on the analyzed model output. The results are easily interpreted in terms of *b*_1_ to *b*_5_, which are associated with the negative potential region and thus with hydrogen bond accepting and basicity properties on the one hand, and *b*_7_ to *b*_10_, which are associated with the positive potential region and thus with hydrogen bond donating and acidity properties on the other hand. *b*_6_ turns out to be related to the neutral potential region insignificantly contributing to the pH value. As for the other *b* coefficient values, the more distant the potential region is from the zero (neutral value), the stronger its influence (whether positive or negative) on the pH value. Thus, the model seems to have a clear and rather simple physical significance. Although statistical analysis showed that the coefficient *b*_6_ was not significant, the variable *S*_6_ was not excluded from the modeling. This result indicates that there is no correlation with the dependent variable at the population level, but this could be changed if a different data set was used.

The ANOVA revealed that the created MLR and PLR models were statistically significant, with *p* values < 0.001. Moreover, higher *F*-test results (*F* value = 39.8120) and lower *p* values, according to Greenland et al. (2016) [20], show the relative relevance of the created models. Based on the presented results it can be concluded that the collected findings demonstrate the dependability of the created models throughout the spectrum of variables evaluated.

### 2.3. Artificial Neural Network Modelling

The applicability of the artificial neural network models for predicting the DES pH values based on the σ profiles was also studied. The best neural network was chosen based on the following criteria: *R*^2^ and *RMSE* for training, test, and validation sets taking into account the number of neurons in the hidden layer. The properties of the created networks that were chosen are shown in Table 3. Based on the goodness of fit and validation error and considering the number of neurons in the hidden layer, the MLP model 10-5-1 was selected as optimal. Fewer neurons in the hidden layer make the ANN architecture simpler. The selected ANN was characterized by 10 neurons in the input layer, 5 neurons in the hidden layer, and 1 neuron in the output layer. The hidden activation function for the selected ANN was Tanh while the output activation function was Logistic. The described ANN provides a good agreement between the experimental data and the data predicted by the model (*R*^2^ _validation_ = 0.9797, *RMSE*_validation_ = 0.0012). As presented in Figure 1c, it can be observed that the data are distributed around the fitted function and that there are no evident outliers. As for the MLP and PLR models, the residual analysis was also performed for the ANN model (Figure 2c) and confirmed the ANN model’s goodness of fit through a normal probability plot of the residuals (Figure 2c), residuals versus the predicted values plot (Figure 2c), histogram of the residuals (Figure 2c), and residuals versus the order of the data plot (Figure 2c).

Based on the presented results, it can be concluded that the σ profiles are good molecular descriptors of DESs since the mathematical correlation of the experimental and predicted values is high. Moreover, based on the obtained *R*^2^ values and the residual analysis, it can be concluded that both the PLR and ANN model can be efficiently applied for the prediction of the DES pH values based on the σ profiles. Due to the simplicity of the PLR model, this model is proposed for the prediction of physicochemical properties.

### 2.4. MLR, PLR, and ANN Models’ Independent Validation

Validation of the MLR, PLR, and ANN models developed for the prediction of the DES pH values based on the σ profiles was performed on the independent set of data. The validation set included the σ profiles of 16 DESs. Comparisons between the experimental data and model-predicted data are shown in Figure 2. The validation performance of the developed models was estimated based on *R*^2^ and *RMSE* and the obtained values were as follows: (i) for MLR *R*^2^ = 0.7097, *RMSE* = 1.1140; (ii) for PLR *R*^2^ = 0.8605, *RMSE* = 0.7652; and (iii) for ANN *R*^2^ = 0.8885, *RMSE* = 0.82926.

It can be noticed that all three proposed models predict the pH value with high accuracy. As expected, the highest *R*^2^ between the experiment and model-predicted data was obtained for ANN prediction of the analyzed DES pH values while the lowest *R*^2^ between the experiment and model-predicted data was obtained for the MLR model. These findings demonstrate that σ profile ANN modeling is a useful and reliable method for predicting DES pH values based on the σ profiles. Nevertheless, considering RMSE, it can be noticed that the PLR model can efficiently be used for the prediction of pH values based on the σ profiles. As described, the *R*^2^ values are scaled between 0 and 1, whereas the RMSE is not scaled to a specific value and, therefore, provides explicit information about how much the prediction deviates.

As stated before, it was relatively easy to link the parameters of the MLR and PLR models to their physical significance. On the other hand, ANNs, by definition, belong to a class of agnostic models and, thus, it is difficult, if not impossible, to reveal their physical meaning. At the same time, this is the reason why they behave much better in interpolation than in extrapolation. The independent validation presented here may be considered as interpolation since the DES members of the independent validation dataset belong to the same DES classes as those used for constructing the model. However, given the rather simple and rather clear relation between the σ profile and pH as revealed by MLR, there is no true reason to believe that the models would behave poorly in extrapolation, even for ANN, i.e., for DES classes not involved in the development of the models. However, this is yet to be checked, e.g., for DESs based on metal chlorides or DESs containing ionic liquids, etc.

The current literature data refer to the prediction of other physicochemical properties (such as viscosity and density) and only a narrow range of values characteristic for limited groups of structurally related DESs [11,12,13,14]. Based on our current knowledge, only one study has investigated the development of a mathematical model for DES pH value prediction [13]. In that study, the pH literature data of 41 DESs were processed in a similar way using the COSMO-RS and mathematical models, MLR and ANN, also covering a variety of cations, anions, and functional groups. The literature study [12] used literature data and included different temperatures (with temperature as an input parameter) while our study used our data obtained at a single temperature. The literature study also showed the potential of MLR and ANN modeling for the prediction of the pH value, however, with more complex models (models with more coefficients) than those developed in this work. Taking into consideration the specific future application of the developed models, it is recommended that they are as simple as possible and as robust as possible. Summing up the presented results, it can be concluded that the PLR model developed in this research can efficiently be used for the prediction of a wide range of DES pH values based on the σ profiles.

## 3. Materials and Methods

### 3.1. Materials

Betaine, choline chloride, glucose, l-(−)-proline, oxalic acid, sucrose, sorbitol, and xylitol were all purchased from Acros Organics, USA. Citric acid, d-fructose, d-(+)-xylose, d,l-malic acid, ethylene glycol, glycerol, and urea were all purchased from Sigma-Aldrich, USA. BIOVIA TmoleX19 version 2021 software (Dassault Systèmes, Vélizy-Villacoublay, France) was used for geometry and energy optimization of the HBAs and HBDs used in this study. BIOVIA COSMOtherm 2020 version 20.0.0. software (Dassault Systèmes) was used for the σ profile calculations of the defined DESs.

### 3.2. Methods

#### 3.2.1. DES Preparation

DESs were prepared by mixing defined molar ratios of HBA to HBD. The two or more components were weighed in a specific ratio in a round-bottomed glass flask, adding 10–50% (*w*/*w*) of water. Then, the flasks were sealed, and the mixtures stirred and heated to 50 °C for 2 h until homogeneous transparent colorless liquids formed. The DES abbreviations and corresponding molar ratios are given in Table 1.

#### 3.2.2. pH Value Measurement

The pH values for each DES were determined with a pH/ion meter S220 using an InLab Viscous Pro-ISM pH-electrode (Mettler Toledo, Greifensee, Switzerland), all within the pH measuring range 0.36–9.31 at room temperature. The instrument was calibrated using standard pH buffer solutions. Additionally, the pH values were checked with litmus paper (range 1–14). All measurements were carried out in duplicates and the results were expressed as an average value ± standard deviation.

#### 3.2.3. Calculation of DES Constituents’ σ Profiles and Descriptors

All molecules used for DES preparation: HBA, HBD, and water, were geometrically and energetically optimized in the BIOVIA TmoleX19 version 2021 (Dassault Systèmes) software. Quantum chemical calculations were performed by adopting DFT (density functional theory) with the BP86 functional level of theory and def-TZVP basis set [10]. To create a simplified and user-friendly database, for each molecule, the single most abundant non-ionized conformer with the lowest energy was chosen and used for further calculations. Molecules consisting of two or more ions (e.g., choline chloride) were treated as ion pairs and their structures were optimized according to Abranches et al. (2019) [21]. Finally, the software-generated COSMO file for each optimized molecule contained its σ profile curve that provided a quantitative representation of the molecules’ polar surface screen charge on the polarity scale. HBAs are characterized by peaks in the negative potential region, HBDs by peaks in the positive potential region, and nonpolar molecules by peaks in the potential region around zero.

To define the molecular descriptors for all DES constituents, the σ profile curve for each HBA, HBD, and water was divided into 10 regions. The width of each region was 0.005 e/Å^2^, covering the range from −0.025 to +0.025 e/Å^2^. The areas under the curve were integrated separately for each defined region. This was achieved by simple summation of the tabulated σ profile data point ordinate values as presented by the BIOVIA COSMOtherm 2020 software. The ordinate values lying on the boundaries of the regions were split into halves and each half was attributed to one of the neighboring regions. Thus, 10 S descriptors (*S*^1^–*S*^10^) of the σ profiles were calculated exactly as the numerical values of these 10 areas (Table A1).

#### 3.2.4. Calculation of DES Descriptors

Any change in the DES composition can be described by a change in its σ profile and the associated numerical value of its descriptors. To obtain a unique descriptor set for each particular DES, the σ profiles of its constituents were processed in the following manner. The descriptors of the studied DESs (*S*^i^_mix_) were calculated from the HBA and HBD component (and in some cases water) descriptors according to Equation (1) proposed by Benguerba et al. (2019) [11]:(1)$$S_{\mathrm{mix}}^{i}=\overset{NC}{\underset{j=1}{\sum }}X_{j}S_{\sigma -\mathrm{profile},j}^{i}$$
where *i* denotes the descriptor number (1–10), *j* stands for the DES constituent number, *X_j_* is the molar fraction of HBA or HBD or some other constituent such as water if present in the mixture, *S*^i^_σ-profile,j_ is the *j*-th constituent *i*-th descriptor, and *NC* is the total number of constituents from which DES is prepared. All the experiments were performed at 20 °C.

#### 3.2.5. Modeling of Correlation between pH and Descriptors

In further calculations, it was assumed that the measured DES pH value can be described as a function of the σ profile of the mixture, expressed by a set of Simix descriptors in Equation (2):(2)$$\mathrm{pH}=f\left(S_{\mathrm{mix}}^{1},S_{\mathrm{mix}}^{2},S_{\mathrm{mix}}^{3},S_{\mathrm{mix}}^{4},S_{\mathrm{mix}}^{5},S_{\mathrm{mix}}^{6},S_{\mathrm{mix}}^{7},S_{\mathrm{mix}}^{8},S_{\mathrm{mix}}^{9},S_{\mathrm{mix}}^{10}\right)$$

Multiple linear regression (MLR) with Equation (3), piecewise linear regression (PLR) with Equation (4), and artificial neural network (ANN) models were attempted to describe the relationship between the input and output variables. The dataset included 142 data points (that included replicates), of which 126 were used for model development and 16 (randomly selected) for independent model validation:(3)$$\mathrm{pH}=b_{0}+b_{1}\cdot S_{\mathrm{mix}}^{1}+b_{2}\cdot S_{\mathrm{mix}}^{2}+b_{3}\cdot S_{\mathrm{mix}}^{3}+b_{4}\cdot S_{\mathrm{mix}}^{4}+b_{5}\cdot S_{\mathrm{mix}}^{5}+b_{6}\cdot S_{\mathrm{mix}}^{6}+b_{7}\cdot S_{\mathrm{mix}}^{7}+b_{8}\cdot S_{\mathrm{mix}}^{8}+b_{9}\cdot S_{\mathrm{mix}}^{9}+b_{10}\cdot S_{\mathrm{mix}}^{10}$$
(4)$$\mathrm{pH}=\left(\left\{\begin{array}{cc} b_{01}+\overset{10}{\underset{i=1}{\sum }}b_{i1}\cdot S_{\mathrm{mix}}^{i} & \forall \left(\mathrm{pH}\leq b_{n}\right)\\ b_{02}+\overset{10}{\underset{i=1}{\sum }}b_{i2}\cdot S_{\mathrm{mix}}^{i} & \forall \left(\mathrm{pH}>b_{n}\right) \end{array}\right\}\right)$$

The PLR technique is based on estimating the parameters of two linear regression equations: one for dependent variable values (*y*) less than or equal to the breakpoint (*bn*) and the other for dependent variable values (*y*) higher than the breakpoint.

The MLR parameters in Equation (3) were estimated using least square regression while the PLR parameters in Equation (4) were estimated using the Levenberg–Marquardt algorithm implemented in the software Statistica 13.0 (Tibco Software Inc, Palo Alto, Santa Clara, CA, USA). The algorithm searches for optimal solutions in the function parameter space using the least squares method. The calculations were performed in 50 repetitions with a convergence parameter of 10–6 and a confidence interval of 95% [22].

In addition, multilayer perceptron (MLP) ANNs were used for the prediction of DES pH values based on the Simix descriptors. The ANN models included an input layer, hidden layer, and output layer. The input layer included 10 neurons representing the Simix descriptors, the output layer had only one neuron, and the number of neurons in the hidden layer varied between 4 and 13 and was randomly selected by the algorithm. The hidden activation function and output activation function were selected randomly from the following set: Identity, Logistic, Hyperbolic tangent, and Exponential. The dimension of the data set for ANN modeling was 126 × 11 and was randomly divided into 70% for network training, 15% for network testing, and 15% for model validation. Model training was carried out using a back error propagation algorithm and the error function was a sum of squares implemented in Statistica v.13.0 Automated Neural Networks. The developed model’s performance was estimated by calculating the R2 and root mean squared error (RMSE) values for the training, test, and validation sets.

Validation of the developed MLR, PLR, and ANN models was performed on an independent data set, including the Simix descriptors for 16 randomly selected DESs. The validation performance of the developed models was estimated based on the R2 and root mean squared error (*RMSE*).

## 4. Conclusions

The applicability of MLR, PLR, and ANN to predict the pH values of DESs was evaluated. The results indicate that although simple linear regression can be used for the description and prediction, its effectiveness and applicability are limited. On the other hand, PLR and ANN are applicable to predict the pH values of DESs with a very high goodness of fit (*R*^2^ > 0.8600). The contribution of this work lies in the development of a user-friendly model to predict pH values in a wide range (from 0.525 to 9.25), indicating that the developed models are good for the prediction of the pH value of newly synthesized DESs. However, due to the simplicity of the developed PLR model, it could be suggested as a model of choice for use in daily work and screening purposes.

Nevertheless, this approach can also be extended to other physicochemical properties since this study confirmed previous findings that showed how the σ profile generated in COSMOtherm is a valuable DES molecular descriptor. It could be a good basis for the evaluation of various mathematical models to develop a simple and applicable prediction model for everyday laboratory or industrial applications.

It is interesting to comment on the influence of the addition of water to a DES. In our previous article [7], based on a limited set of data, it was noticed that the addition of water to extremely acidic DESs increases their pH values, and the addition of water to highly basic DESs decreases their pH values. Thus, it seemed that the addition of water somehow mellowed the pH environments. On the other hand, on a larger set of data, as presented here, this conclusion does not hold any more: there are difficult-to-predict exemptions to the rule. On the other hand, the COSMO-RS calculation results in combination with the non-presumptive numerical models, such as MLR, PLR, and ANN, are perfectly suitable to tackle those difficult-to-predict systems.

## Figures and Tables

**Figure 1 molecules-27-04489-f001:**
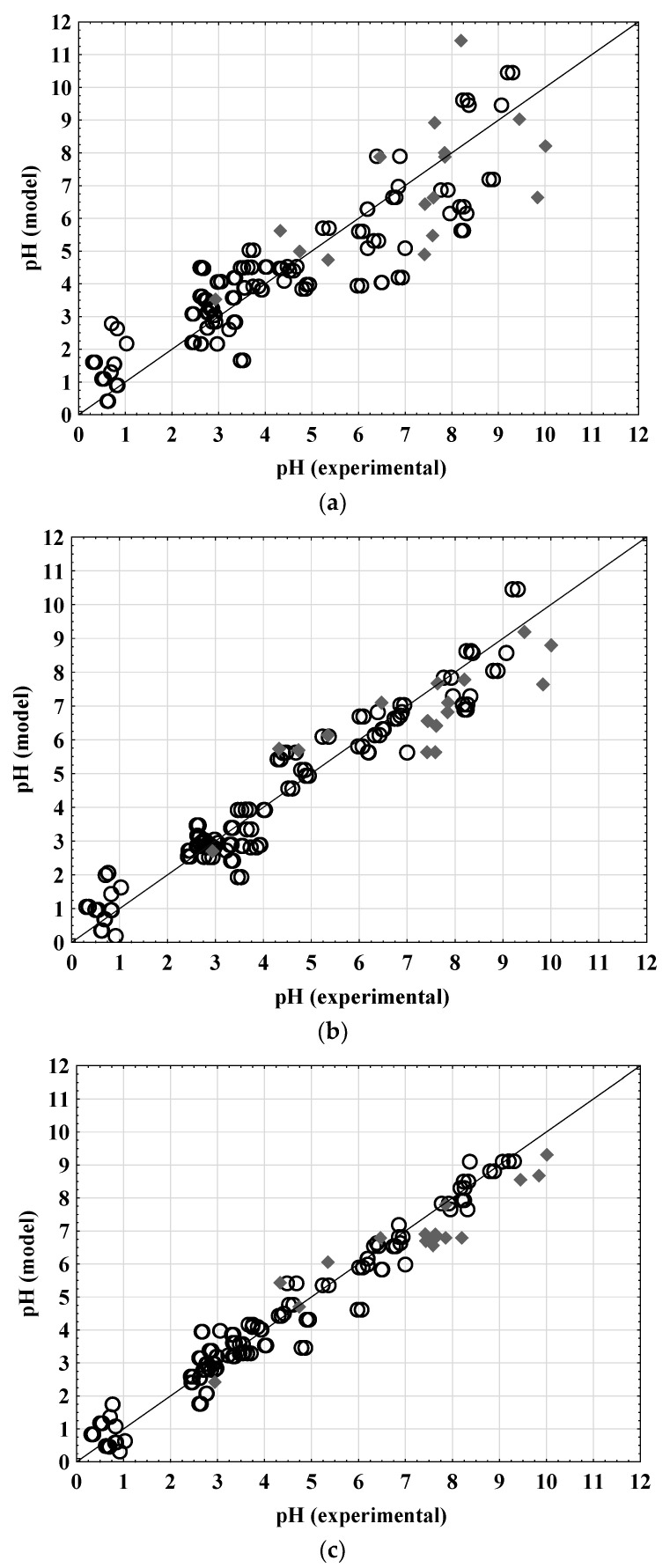
Comparison between experimental data and (**a**) MLR model, (**b**) PLR model, and (**c**) ANN model. (○) data set for model development, (◆) data set for model validation.

**Figure 2 molecules-27-04489-f002:**
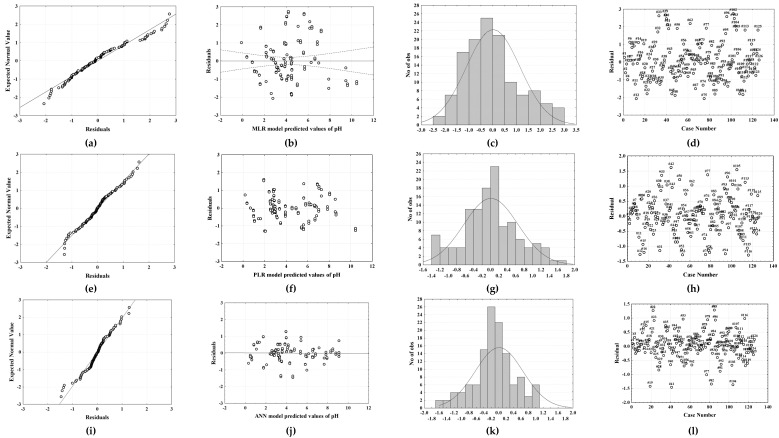
Analysis of the residuals for the MLR model (**a**–**d**), PLR model (**e**–**h**), and ANN mode (**i**–**l**).

**Table 1 molecules-27-04489-t001:** Experimentally measured pH values.

DES	Abbreviation	Molar Ratio	*w*H_2_O [%]	pH (20 °C) ± st.dev.
Betaine:citric acid	B:CA	1:1	30	2.46 ± 0.04
			50	2.46 ± 0.02
Betaine:ethylene glycol	B:EG	1:2	30	6.86 ± 0.00
Betaine:glucose	B:Glc	1:1	10	6.64 ± 0.35
Betaine:glycerol	B:Gly	1:2	30	6.77 ± 0.04
			50	6.38 ± 0.07
Betaine:oxalic acid:glycerol	B:OxA:Gly	1:2:1	30	2.91 ± 0.05
Betaine:malic acid	B:Ma	1:1	30	2.98 ± 0.01
			50	2.92 ± 0.01
Betaine:sucrose	B:Suc	4:1	30	7.85 ± 0.11
Choline chloride:citric acid	Ch:CA	2:1	30	0.34 ± 0.04
			50	0.71 ± 0.00
Choline chloride:ethylene glycol	ChCl:EG	1:2	10	6.19 ± 0.01
			30	6.60 ± 0.57
			50	4.58 ± 0.14
			80	4.41 ± 0.00
Choline chloride:fructose	ChCl:Fru	1:1	30	3.51 ± 0.05
			50	3.35 ± 0.03
Choline chloride:glucose	ChCl:Glc	1:1	30	4.83 ± 0.06
			50	3.56 ± 0.01
Choline chloride:glycerol	ChCl:Gly	1:2	30	3.71 ± 0.06
			50	2.67 ± 0.11
			80	3.06 ± 0.01
Choline chloride:malic acid	ChCl:MA	1:1	30	0.63 ± 0.01
			50	1.03 ± 0.00
Choline chloride:proline:malic acid	ChCl:Pro:MA	1:1:1	10	3.23 ± 0.00
			30	2.82 ± 0.01
			50	2.63 ± 0.03
Choline chloride:sorbitol	ChCl:Sol	1:1	50	4.92 ± 0.04
			80	3.80 ± 0.08
Choline chloride:urea	ChCl:U	1:2	10	9.26 ± 0.08
			30	8.85 ± 0.06
			50	8.23 ± 0.04
Choline chloride:urea:ethylene glycol	ChCl:U:EG	1:2:2	10	8.29 ± 0.07
Choline chloride:urea:glycerol	ChCl:U:Gly	1:2:2	10	8.72 ± 0.05
Choline chloride:xylose	ChCl:Xyl	2:1	30	2.86 ± 0.04
			50	3.32 ± 0.03
			80	3.93 ± 0.01
Choline chloride:xylitol	ChCl:Xyol	5:2	30	6.90 ± 0.06
			50	6.50 ± 0.01
			80	6.03 ± 0.06
Choline chloride:fructose	ChCl:Fru	1:1	30	3.51 ± 0.05
			50	3.35 ± 0.03
Citric acid:glucose	CA:Glc	1:1	30	0.53 ± 0.04
Citric acid:sucrose	CA:Suc	1:1	30	0.83 ± 0.00
Fructose:ethylene glycol	Fru:EG	1:2	30	5.31 ± 0.09
Fructose:glucose:ethylene glycol	Fru:Glc:EG	1:1:2	50	3.67 ± 0.06
Fructose:glucose:sucrose	Fru:Glc:Suc	1:1:1	50	2.63 ± 0.03
			80	2.99 ± 0.01
Fructose:glucose:urea	Fru:Glc:U	1:1	30	8.22 ± 0.06
Glucose:ethylene glycol	Glc:EG	1:2	50	4.03 ± 0.02
Glucose:glycerol	Glc:Gly	1:2	50	4.33 ± 0.04
Malic acid:fructose	MA:Fru	1:1	30	0.77 ± 0.01
Malic acid:fructose:glycerol	MA:Fru:Gly	1:1	30	2.77 ± 0.01
Malic acid:glucose	MA:Glc	1:1	30	0.83 ± 0.01
Malic acid:glucose:glycerol	MA:Glc:Gly	1:1:1	10	0.92 ± 0.00
Malic acid:sucrose	MA:Suc	2:1	30	0.66 ± 0.01
Proline:malic acid	Pro:MA	1:1	10	2.63 ± 0.01
			30	2.78 ± 0.02
			50	2.73 ± 0.03
Sucrose:ethylene glycol	Suc:EG	1:2	30	6.05 ± 0.06
Sucrose:glucose:urea	Suc:Glc:U	1:1	30	8.14 ± 0.25
Xylose:ethylene glycol	Xyl:EG	1:2	30	4.57 ± 0.06

**Table 2 molecules-27-04489-t002:** MLR and PLR regression coefficients. Statistically significant coefficients are marked in bold.

	MLR		PLR	
	Regression Coeff. ± st. Error	*p*-Value	Regression Coeff. ± st. Error	*p*-Value
Break point			**4.1246 ± 0.3292**	0.0021
*b* _0_	**−13.4623 ± 4.9782**	0.0078	**−1.9449 ± 0.1556 −80.4560 ± 10.6436**	0.0001
*b*_1_ (*S*^1^_mix_)	**16.4623 ± 5.1388**	0.0022	**14.8847 ± 2.1908 −23.1982 ± 1.8558**	0.0001
*b*_2_ (*S*^2^_mix_)	**9.1349 ± 2.4418**	0.0003	**10.2415 ± 2.3918** **27.8095 ± 2.2247**	0.0001
*b*_3_ (*S*^3^_mix_)	**9.7560 ± 2.5748**	0.0002	**9.1933 ± 1.7354** **35.1992 ± 2.8159**	<0.0001
*b*_4_ (*S*^4^_mix_)	**4.2440 ± 1.1602**	0.0004	**4.8581 ± 1.1221** **11.2879 ± 1.1902**	<0.0001
*b*_5_ (*S*^5^_mix_)	**2.2980 ± 0.6482**	0.0006	**2.5621 ± 0.1188** **10.1747 ± 1.3976**	<0.0001
*b*_6_ (*S*^6^_mix_)	−0.9176 ± 1.0696	0.3927	−2.4281 ± 0.8779 −14.7126 ± 1.1770	0.2666
*b*_7_ (*S*^7^_mix_)	**−4.5381 ± 1.1435**	0.0020	**−4.1497 ± 0.6632** **−9.6777 ± 0.7742**	<0.0001
*b*_8_ (*S*^8^_mix_)	**−8.9573 ± 1.9634**	<0.0001	**−9.2237 ± 1.6373 −25.6581 ± 2.0526**	<0.0001
*b*_9_ (*S*^9^_mix_)	**−10.0312 ± 2.8589**	0.0006	**−11.4736 ± 3.6473 −32.0013 ± 2.5601**	0.0001
*b*_10_ (*S*^10^_mix_)	**−12.9604 ± 3.6943**	0.0006	**−13.9250 ± 4.4560 −42.7492 ± 3.4199**	0.0001
*R* ^2^	0.7758		0.9654	
*R* ^2^ _adj_	0.7564		0.9624	
*RMSE*	1.1865		0.6558	
*F* value	39.8120		39.8120	
*p*-value	<0.0001		<0.0001	

**Table 3 molecules-27-04489-t003:** Architecture of the developed ANN (selected network is marked in bold). The numbers in the network name denote the number of neurons in the input, hidden, and output layers, respectively.

Network Name	Training Perf./ Training Error	Test Perf./ Test Error	Validation Perf./ Validation Error	Hidden Activation	Output Activation
MLP 10-13-1	0.9734, 0.0021	0.9751, 0.0031	0.9578, 0.0042	Logistic	Logistic
MLP 10-11-1	0.9812, 0.0013	0.9802, 0.0018	0.9794, 0.0018	Tanh	Exponential
MLP 10-10-1	0.9803, 0.0013	0.9827, 0.0016	0.9788, 0.0019	Tanh	Tanh
MLP 10-10-1	0.9808, 0.0017	0.9806, 0.0021	0.9716, 0.0019	Tanh	Logistic
**MLP 10-5-1**	**0.9868, 0.0011**	**0.9799, 0.0012**	**0.9797, 0.0012**	**Tanh**	**Logistic**

## Data Availability

Not applicable.

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
