# Peer review of "Prediction of pH Value of Aqueous Acidic and Basic Deep Eutectic Solvent Using COSMO-RS σ Profiles’ Molecular Descriptors"

_molecules, 2022, doi:10.3390/molecules27144489_

Round 1

Reviewer 1 Report

This study has developed a simple and easy-to-apply model to predict the pH-values of the deep eutectic solvents (DES) from a wide range of pH-values, which evaluate the applicability of the pH-values of the DES in multiple linear regression (MLR), piecewise linear regression (PLR), and artificial neural networks (ANN). Results show that PLR and ANN are suitable for predicting the pH-values of DES, and the goodness of fit is very high (R2> 0.8600). Due to the simplicity of the PLR model developed, it can be used for daily work.

1.On the line 153, "A4" should be changed to "a4".

2.The color of the figures is relatively single.

3.On the line 249, "DESs" should be changed to "DES", it should be unified with above.

4. DOI errors in the papers in References 9 and 14.

5. The typographical distribution of formula(4) is confusing.

6. The first letter is not capitalized on page 4 (line 315).

7. The legend is missing in Figure 2.

8. The title sequence number is wrong. The title sequence numbers after 3.2.3 in the article are "4.2.4" and "4.2.5" respectively, and the first letters of all words are not capitalized in these two titles, which is inconsistent with the title format of the same level .

9. In Table 1, the "H2O" in the header should be "H2O".

10. The a, b, c, d, e, and f mentioned in the legend are not clearly marked in the 12 pictures of Figure 2.

Author Response

RESPONSE TO REVIEWERS

We would first like to thank the Reviewers for their efforts and detailed comments of our manuscript “Prediction of pH-value of aqueous acidic and basic deep eutectic solvent using COSMO-RS σ-profiles molecular descriptors”. A complete list of comments and our responses is given below.

Reviewer #1: This study has developed a simple and easy-to-apply model to predict the pH-values of the deep eutectic solvents (DES) from a wide range of pH-values, which evaluate the applicability of the pH-values of the DES in multiple linear regression (MLR), piecewise linear regression (PLR), and artificial neural networks (ANN). Results show that PLR and ANN are suitable for predicting the pH-values of DES, and the goodness of fit is very high (R2> 0.8600). Due to the simplicity of the PLR model developed, it can be used for daily work.

Comment #1.On the line 153, "A4" should be changed to "a4".

Response: Revised as suggested.

Comment #2.     The color of the figures is relatively single.

Response: Color is to be used only when truly necessary as it make publication more expensive. We think that everything important is clearly seen with the black and white figures as presented.

Comment #3.    On the line 249, "DESs" should be changed to "DES", it should be unified with above.

Response: Revised as suggested.

Comment #4. DOI errors in the papers in References 9 and 14.

Response: Revised as suggested.

Comment #5. The typographical distribution of formula(4) is confusing.

Response: The reviewer is right. Formulas may be written more simple, as follows:

(WRITTEN IN PDF VERSION OF REPORT)

Formula 4 was not correct. It may be written correctly, and more simple, as follows:

(WRITTEN IN PDF VERSION OF REPORT)

Comment #6.  The first letter is not capitalized on page 4 (line 315).

Response: Revised as suggested.

Comment #7.  The legend is missing in Figure 2.

Response: Revised as suggested.

Comment #8.  The title sequence number is wrong. The title sequence numbers after 3.2.3 in the article are "4.2.4" and "4.2.5" respectively, and the first letters of all words are not capitalized in these two titles, which is inconsistent with the title format of the same level .

Response: Revised as suggested.

Comment #9.   In Table 1, the "H2O" in the header should be "H2O".

Response: Revised as suggested.

Comment #9.   The a, b, c, d, e, and f mentioned in the legend are not clearly marked in the 12 pictures of Figure 2.

Response: Revised as suggested.

Reviewer 2 Report

This manuscript presents results of a research performed for finding a way to predict pH-value of aqueous acidic and basic deep eutectic solvent using COSMO-RS σ-profiles molecular descriptors. This study is of value and can be accepted in your journal after some revision stated below:

(I) 3.2.2. pH-Value Measurement: while the pH-meter was calibrated in the aqueous media, is it possible to correctly get pH values in the other media? The same for litmus paper.

(II) Conclusion: please add a few lines about how water influences the measured pH-value. As mentioned in line 60, amount of water may greatly affect the pH of DES; in the other words, a particular DES with different water content will have a different pH, which is not clearly explained in this article.

(III) Table 1: please state the reason of selecting these particular DESs.

Author Response

RESPONSE TO REVIEWERS

We would first like to thank the Reviewers for their efforts and detailed comments of our manuscript “Prediction of pH-value of aqueous acidic and basic deep eutectic solvent using COSMO-RS σ-profiles molecular descriptors”. A complete list of comments and our responses is given below.

Reviewer #2:  

This manuscript presents results of a research performed for finding a way to predict pH-value of aqueous acidic and basic deep eutectic solvent using COSMO-RS σ-profiles molecular descriptors. This study is of value and can be accepted in your journal after some revision stated below:

Comment #1. 3.2.2. pH-Value Measurement: while the pH-meter was calibrated in the aqueous media, is it possible to correctly get pH values in the other media? The same for litmus paper.

Response: The same electrode (InLab Viscous Pro-ISM pH-electrode) for all solvents is used. The instrument was calibrated using standard pH buffer solutions. Tested solvents are water solutions so we think that experiment was performed correctly. On the other hand, the reviewer is right. Strictly speaking, pH is correctly defined in aqueous solutions only. By changing the medium, the values that are measured are on fact the instrument responses and not the classically defined pH values. However, they do indicate the basicity or acidity of the medium which is the main purpose of the experiment presented.

Comment #2. Conclusion: please add a few lines about how water influences the measured pH-value. As mentioned in line 60, amount of water may greatly affect the pH of DES; in the other words, a particular DES with different water content will have a different pH, which is not clearly explained in this article.

Response: The following text is added:

It is interesting to comment on the influence of adding water to a DES. In our previous article[1], based on a limited set of data, it was noticed that adding water to extremely acidic DESs increase their pH values, and adding water to highly basic DESs decreases their pH values. Thus it seemed that adding water somehow mellowed the pH environments. On the other hand, on a larger set of data as presented here, this conclusion does not hold any more: there are difficult-to-predict exemptions to the rule. On the other hand, COSMO-RS calculation results in combination with non-presumptive numerical models, such as MLR, PLR and ANN are perfectly suitable to tackle those difficult-to-predict systems.

Comment #1. Table 1: please state the reason of selecting these particular DESs.

Response: We used DESs for our database. In particular, we wanted to develop a user-friendly model to predict pH values in the wide range and for this reason, we selected both acidic and basic DES (pH values from 0.525 to 9.25). (L96-97)

[1] Mitar, A.; Panić, M.; Prlić Kardum, J.; Halambek, J.; Sander, A.; Zagajski Kučan, K.; Radojčić Redovniković, I.; Radošević, K. Physicochemical Properties, Cytotoxicity, and Antioxidative Activity of Natural Deep Eutectic Solvents Containing Organic Acid. Chemical and Biochemical Engineering Quarterly 2019, 33, 1–18, doi:10.15255/CABEQ.2018.1454.

Round 2

Reviewer 2 Report

Can be accepted, properly revised.